# Focal Cryo-Immunotherapy with Intratumoral IL-12 Prevents Recurrence of Large Murine Tumors

**DOI:** 10.3390/cancers15082210

**Published:** 2023-04-08

**Authors:** Maura R. Vrabel, Jacob A. Schulman, Francis B. Gillam, Siena M. Mantooth, Khue G. Nguyen, David A. Zaharoff

**Affiliations:** 1ImmunoEngineering Laboratory, Joint Department of Biomedical Engineering, UNC-Chapel Hill and NC State University, Raleigh, NC 27695, USA; 2Comparative Medicine Institute, NC State University, Raleigh, NC 27695, USA; 3Lineberger Comprehensive Cancer Center, UNC-Chapel Hill, Chapel Hill, NC 27599, USA

**Keywords:** focal ablation, cryoablation, IL-12, intratumoral, immunotherapy, abscopal response, checkpoint therapy

## Abstract

**Simple Summary:**

Focal cryoablation is an FDA-approved minimally-invasive treatment for certain kidney, colon, lung, breast, and prostate cancers. However, it is difficult to ensure complete tumor destruction after ablation, and therefore, tumor recurrence rates can be as high as 80%. In order to improve primary tumor regression, we explored the addition of an immunotherapy, interleukin-12 (IL-12), delivered intratumorally, to cryoablation. We also explored the impact of this cryo-immunotherapy on systemic antitumor immunity using bilateral tumors and spontaneous metastasis models. The combination of cryoablation and IL-12 resulted in the durable regression of large (>200 mm^3^) primary tumors in multiple models and reduced lung metastasis when IL-12 was delivered as a neoadjuvant. Possible mechanisms for this anti-tumor effect were explored using whole transcriptome RNAseq.

**Abstract:**

Focal ablation technologies are routinely used in the clinical management of inoperable solid tumors but they often result in incomplete ablations leading to high recurrence rates. Adjuvant therapies, capable of safely eliminating residual tumor cells, are therefore of great clinical interest. Interleukin-12 (IL-12) is a potent antitumor cytokine that can be localized intratumorally through coformulation with viscous biopolymers, including chitosan (CS) solutions. The objective of this research was to determine if localized immunotherapy with a CS/IL-12 formulation could prevent tumor recurrence after cryoablation (CA). Tumor recurrence and overall survival rates were assessed. Systemic immunity was evaluated in spontaneously metastatic and bilateral tumor models. Temporal bulk RNA sequencing was performed on tumor and draining lymph node (dLN) samples. In multiple murine tumor models, the addition of CS/IL-12 to CA reduced recurrence rates by 30–55%. Altogether, this cryo-immunotherapy induced complete durable regression of large tumors in 80–100% of treated animals. Additionally, CS/IL-12 prevented lung metastases when delivered as a neoadjuvant to CA. However, CA plus CS/IL-12 had minimal antitumor activity against established, untreated abscopal tumors. Adjuvant anti-PD-1 therapy delayed the growth of abscopal tumors. Transcriptome analyses revealed early immunological changes in the dLN, followed by a significant increase in gene expression associated with immune suppression and regulation. Cryo-immunotherapy with localized CS/IL-12 reduces recurrences and enhances the elimination of large primary tumors. This focal combination therapy also induces significant but limited systemic antitumor immunity.

## 1. Introduction

A significant proportion of total cancer-related mortality is due to inoperable, or unresectable, solid malignancies. Despite improvements in cancer imaging and detection, relatively large percentages of solid tumors are found to be inoperable at the time of diagnosis, either due to the proximity of the tumor to major organs and vasculature or due to patient comorbidities. For example, approximately 80% of cholangiocarcinomas [1], 80–85% of pancreas tumors [2,3], 70% of liver tumors [4], >50% of esophagus tumors [5], 40% of lung tumors [6], and even 10–15% of breast cancers [7] are unresectable at the time of initial diagnosis. For these patients, minimally-invasive tumor ablation treatments are essential.

Tumor ablation technologies, which include radiofrequency ablation (RFA), microwave ablation (MWA), high intensity focused ultrasound (HIFU), and cryoablation (CA), utilize diverse forms of energy and physical mechanisms to directly kill tumor tissues. Several reviews comparing different tumor ablation strategies have been published recently [8,9,10,11,12]. CA is an established method for treating certain prostate cancers [13,14,15] and small renal masses [16,17,18]. It is also a treatment option for inoperable lung, liver, bone, eye and cervical cancers. CA is still considered investigational for breast cancer, although it is routinely used to freeze benign fibroadenomas.

Although CA is effective in immediately destroying large tumors, negative margins are not always achievable. Residual cancer deposits following CA drive higher rates of local recurrence compared to surgical resection. CA of hepatocellular carcinomas (HCCs) results in 20% local and 60% distant recurrences [19]. Against the small T1a renal cell carcinomas, clinical outcomes following CA and partial nephrectomy are similar [20,21,22,23,24,25,26]. However, when renal tumors exceed 4 cm, CA results in inferior recurrence rates [27,28]. Large inoperable tumors would benefit from an adjuvant therapy capable of eliminating residual cancer cells and thus preventing recurrences.

The goal of the current study was to determine if localized immunotherapy could reduce recurrence rates following CA of large, established tumors. Initial studies compared the efficacies of multiple immune adjuvants applied intratumorally following CA. Follow-up experiments explored the ability of CS/IL-12, a formulation of IL-12 in a chitosan (CS) biopolymer solution, to inhibit recurrences after CA in multiple primary murine tumor models. We have previously shown that viscous solutions of CS can maintain high levels of coformulated proteins, including IL-12, in an injected tumor for at least one week [29,30,31]. Immunotherapy combination studies in bilateral and metastatic tumor models investigated the generation of systemic antitumor immunity following focal ablation treatments. Bulk RNAseq analyses offered a first glimpse into the potential mechanisms by which CS/IL-12 improves CA outcomes. Data presented provide a strong rationale for combining focal ablation with potent localized immunotherapies for improved outcomes when treating inoperable tumors.

## 2. Materials and Methods

### 2.1. Animals and Cell Lines

Female C57BL/6 mice, aged 6–12 weeks old, were obtained from the Charles River Laboratories and maintained in a pathogen-free facility. Animal use complied with the Public Health Service Policy on Human Care and Use of Laboratory Animals. All experiments involving laboratory animals were approved by the Institutional Animal Care and Use Committee at North Carolina State University. MB49, MC38 and Panc02 cell lines were kindly shared by Dr. Jeffrey Schlom, Laboratory of Tumor Immunology and Biology, National Cancer Institute. LLC1 (LL/2) and B16.F10 cells were obtained from ATCC (Manassas, VA, USA). Cells were cultured at a low passage number and prepared in Dulbecco’s phosphate buffered saline (DPBS) without Ca^2+^ or Mg^2+^ for implantation. For detailed media recipes, see Appendix A.

Subcutaneous tumor implantations were performed under O_2_/isoflurane anesthesia to ensure accurate placement. To establish primary tumors, 3 × 10^5^ cells (B16.F10, MC38 and MB49) or 1 × 10^6^ cells (Panc02) were injected subcutaneously (s.c.) on the right flank in either 100 µL or 50 µL of DPBS. Tumors were measured with digital calipers and tumor volumes were calculated using the formula ½*A*B^2^, where A and B are the perpendicular long and short dimensions, respectively. Mice were randomized into treatment groups based on average tumor volume and tumors were treated at volumes measuring between 200 and 500 mm^3^. Only tumor volume was considered for the randomization criteria. Unless otherwise stated, CA was performed percutaneously using the Visual-ICE^TM^ Cryoablation System (Boston Scientific, Marlborough, MA, USA) in three freeze/thaw cycles with 1 min at 100% freezing intensity and then 2 min of active thaw, where the duration of the last thaw was 1 min. In all instances, the ice ball encapsulated the entire tumor. The CA needle was custom-made at 18 gauge and 1.5 inches for small rodent procedures. Mice were anesthetized with the continuous flow of isoflurane for the duration of the cryoablation treatment. The threshold to euthanize mice was a tumor burden of 2000 mm^3^.

For bilateral models (MC38 and MB49), 3 × 10^5^ cells were implanted s.c. in the right flank and 1.5 × 10^5^ cells were implanted in the left flank on the same day. For the bilateral Panc02 model, 1 × 10^6^ cells were implanted on the right flank and 0.5 × 10^6^ cells were implanted on the left flank. The larger of the two tumors was cryoablated at a volume measuring between 200 and 400 mm^3^. The euthanasia threshold was a total tumor burden of 2000 mm^3^.

For the spontaneous metastasis model, 5 × 10^5^ LLC1 cells were injected s.c. on the right flank and tumors measuring between 200 and 500 mm^3^ were either resected or treated with CA. For the resection surgery, mice were anesthetized intraperitoneally (i.p.) with a solution of ketamine (75 mg/kg) and xylazine (15 mg/kg) in sterile water. The surgical area was shaved, then disinfected using betadine solution followed by a 70% ethanol solution. The primary tumor was then excised by lifting the tumor with forceps and cut around the base using a surgical blade. Blood vessels feeding the primary tumor were cauterized and the resection site was closed with wound clips. Mice were euthanized 21 days after CA and lungs were dissected to evaluate the metastatic burden via tumor nodule enumeration. Gross pathology was performed on dissected lungs to identify lung metastases. Extracted lungs were observed under a stereomicroscope. Metastatic nodules were identified as small round masses protruding from normal lung tissue and were counted manually.

### 2.2. Immunotherapy

Recombinant murine IL-12 was overexpressed and purified, as described previously [32]. IL-12 was formulated in 1.5 *w*/*v*% chitosan acetate (CS) (Heppe Medical Chitosan, Halle, Germany) dissolved in DPBS, as described previously [33]. Unless otherwise noted, the dose of IL-12 was 1 µg and CS/IL-12 was injected intratumorally within an hour of CA. R848 (Resiquimod) (Invivogen, San Diego, CA, USA) was formulated in PBS and delivered intratumorally at 10 µg/mouse. DMXAA (Invivogen) was formulated in DMSO and delivered intratumorally at 500 µg/mouse. For the anti-PD-1 and isotype antibody (BioXCell, Lebanon, NH, USA, clone: RMP1.14) treatments, 300 µg was injected i.p. every 3 days, starting on the day of CA for a total of 6 doses.

### 2.3. Transcriptome Sequencing

MC38 tumors were implanted subcutaneously in the right flanks of C57BL/6 mice, as described above. When tumor volumes measured between 200 and 500 mm^3^, CA was performed and CS/IL-12 was intratumorally injected within an hour of CA. Mice were euthanized 3 days and 7 days post ablation to harvest tumor tissues and tumor draining inguinal lymph nodes (dLNs). To obtain cell suspensions, the dLNs were homogenized through 40 µm mesh strainers with complete T cell media (Appendix A). Tumor tissues were first minced into approximately 0.5 cm squares before treatment with a 5X Triple Enzyme Solution (collagenase type IV 10 mg/mL, hyaluronidase 1 mg/mL, and DNase 20,000 units final concentration in Hank’s Buffered Saline Solution) for 30 min at 37 °C with magnetic bar stirring. The resulting homogenate was filtered through a 70 µm mesh strainer into a clean tube to achieve the final cell suspension.

Total RNA was extracted from the cell suspension using the GeneJET RNA Purification Kit (ThermoFisher cat #K0731, Waltham, MA, USA), according to the manufacturer’s protocol. Briefly, Lysis Buffer, supplemented with β-mercaptoethanol and then ethanol (99%), was added to the cell suspension and vortexed. The lysate was then loaded onto the purification spin column and washed twice before eluting with nuclease-free water. Samples were sent to BGI Genomics (Hong Kong, China) to perform transcriptome sequencing using their DNBSEQ^TM^ platform with a read length of 100 paired-end base pairs. Data analysis was performed using the BGI software platform, Dr. Tom.

Sample inclusion was based on RNA quality (260/280 > 1.8) and successful library construction. Only one sample, from a tumor treated with CA + CS/IL-12 (11_T_D3), failed to meet these criteria and was omitted from further analysis. The dLN from one mouse treated with CA alone and then euthanized on day 7 after treatment was not recovered during necropsy (34_dLN_D7). Secondary exclusion criteria were based on read alignment coverage and randomness. The technical difficulty of extracting high quality RNA from highly necrotic ablated tumor samples, particularly three days after ablation, hindered the interpretation of gene expression from the tumor at day 3 post treatment.

Unless otherwise noted, for all gene expression analysis, genes were filtered from the transcriptome using a gate of Q < 0.05 for the comparisons of interest. Bowtie2 was used to map the clean reads to the reference gene sequence (transcriptome), and then RSEM was used to calculate the gene expression level of each sample. Differential gene detection was calculated using the DESeq2 method, where Q is calculated using the FDR/Benjamini–Hochberg method. According to the results of differential gene detection, the R package pheatmap was used to perform hierarchical clustering analysis on the union set differential genes.

Gene expression data that have been cleaned (QC and Filter) and aligned to the reference genome have been deposited on the Dryad Data Platform and are available at: https://doi.org/10.5061/dryad.g4f4qrfts.

### 2.4. Statistics

Statistical analysis, including checks for correct model, equal variance, and normality, was performed using GraphPad Prism 9.5.0.730. Student’s *t*-test was performed for comparisons of two groups. One-way analysis of variance (ANOVA) with Tukey’s post-test correction for multiple comparisons was performed for comparisons of three of more groups. For survival analyses, the Log-rank (Mantel–Cox) test was used and individual pairwise comparisons were made. Significance was determined using the Bonferroni corrected significance level (α_B_), where α_B_ = α/k and k is the number of comparisons of interest. All analyses assumed the Gaussian distribution, equal standard deviations, and were unpaired.

## 3. Results

### 3.1. Localized IL-12, but Not TLR or STING Agonists, Improves Primary Tumor Elimination and Prevents Recurrence after Ablation in Multiple Tumor Models

To examine the therapeutic effect of different immune adjuvants to CA, large (200–500 mm^3^) murine tumors were established subcutaneously on the flank and treated with either CA alone or CA plus intratumoral immunotherapy. Adjuvant CS/IL-12 achieved the greatest tumor-free survival at 90% (Figure 1A). The addition of Resiquimod (R848), a TLR7/8 agonist, prevented tumor recurrence in 60% of mice while adjuvant DMXAA did not enhance long-term survival compared to CA alone, both resulting in an overall survival of 40% (Figure 1A). A comparison of CS/IL-12 and R848 treatments, though not statistically significant at *p* = 0.1359, demonstrates a crucial trend of improved survival rate with CS/IL-12. Further comparisons with DMXAA and CA alone reach statistical significance at the family-wise significance level of 0.05, with *p* values of 0.0253 and 0.0269, respectively. It is important to note, however, that these *p* values fall in the non-significant category when a stricter interpretation is applied with a significance level of 0.0167 resulting from the Bonferroni correction for multiple comparisons (Figure 1A).

Additional models were explored to confirm the trends noted in the MB49 model (Figure 1A) and to assess if outcomes were tumor dependent. Mice, bearing either MC38 or Panc02 heterotopic tumors, experienced similar survival increases when CS/IL-12 was added following CA. In each model, 40% of mice experienced a progressive recurrence following CA alone, while CA + CS/IL-12 prevented recurrences in all treated mice (Figure 1B,C) leading to 100% long-term survival. In the non-immunogenic B16.F10 model, 67% of mice developed local, and lethal recurrences following CA alone. The recurrence rate was reduced to 20% with the addition of CS/IL-12 (Figure 1D). Eighty percent of mice bearing large B16.F10 tumors remained tumor-free for more than 40 days after cryo-immunotherapy with CA + CS/IL-12. Again, while comparisons are not strictly statistically significant using the 0.05 significance level, there is strong evidence for the claim that the addition of CS/IL-12 to CA greatly improves overall survival and decreases primary tumor recurrence in not just one, but four different tumor models with varying immunogenicities.

### 3.2. CS/IL-12 Delivered within 4 Days after CA Is the Most Effective Therapeutic Window

In order to understand the temporal dependence of CS/IL-12 delivery on recurrence and survival outcomes, large MB49 tumors were treated with CS/IL-12 delivered intratumorally 0, 2, 4, or 6 days after CA, where day 0 refers to delivery within an hour of ablation. Overall survival for day 0, 2 and 4 treatments were similar and was between 75% and 87.5% (Figure 2A). The differences between overall survival at day 0, 2, and 4 are not statistically significant at the family-wise significance level of 0.05, nor at the Bonferroni corrected threshold of 0.003. However, waiting 6 days after CA to administer CS/IL-12 drops the overall survival to 50%. Although this drop is not statistically significant compared to day 0 (*p* = 0.1061), it demonstrates a trend of weakening efficacy when CS/IL-12 is delivered long after the initial ablation. To further support the claim for earlier intratumoral CS/IL-12 delivery after CA, waiting 6 days did not significantly enhance overall survival compared to CA alone at 45% (*p* = 0.5122). In contrast, day 0 CS/IL-12 compared to CA alone was statistically significant at the family-wise significance level of 0.05, where *p* = 0.0431 (Figure 2A).

Associated tumor volume data gives a clearer picture of individual outcomes, where 7 out of 8 mice have durable regressions after delivery of CS/IL-12 on the same day as CA (Figure 2B). Immunotherapy delivery on day 2 or day 4 yields similar results with 6 out of 8 and 6 out of 7 mice experiencing complete regression, respectively. The day 6 CS/IL-12 treatment group resulted in only 5 out of 10 mice experiencing recurrence-free regressions. Furthermore, CA without adjuvant immunotherapy results in 3 out of 8 mice with durable regression (Figure 2B). Altogether these data demonstrate the high frequency of recurrences following CA alone and the high frequency of durable cures when adjuvant CS/IL-12 is administered within 4 days (Figure 2B).

### 3.3. Addition of CS/IL-12 to CA Inhibits Lung Metastases but Not Established Abscopal Tumors

Due to the significant local tumor response after CA + CS/IL-12, we sought to explore systemic antitumor effects. The spontaneously metastatic LLC1 model was used to determine if CS/IL-12 immunotherapy could inhibit metastases following CA. Preclinical [34,35] and clinical [36,37,38] data have demonstrated improved oncological outcomes after neoadjuvant delivery of immunotherapy. Therefore, CS/IL-12 was delivered intratumorally two days before CA in order to assess the metastatic response to neoadjuvant immunotherapy. Mice were observed for 21 days after intervention and then sacrificed to examine lung metastases. In regards to primary tumor recurrence, there was no statistical significance between the two treatment groups, where 5 of 9 mice experienced a primary tumor recurrence in the CS/IL-12 plus CA group, compared to 7 of 9 mice for CA alone (Figure 3A,B). However, the neoadjuvant addition of CS/IL-12 to CA dramatically reduced the number of metastatic lung nodules compared to CA alone, from 12 to 0.4 metastatic nodules on average, thus demonstrating a 97% reduction (Figure 3C).

To compare these cryo-immunotherapy results with a standard treat and resect model, LLC1 tumors were treated intratumorally with neoadjuvant CS/IL-12 before resection or simply resected without immunotherapy. Primary tumor recurrence was completely prevented with the addition of neoadjuvant CS/IL-12, while 87.5% of mice who underwent only tumor resection experienced a recurrence. This enhancement in protection from primary tumor recurrence was statistically significant (Figure 3D,E). Similar to the results using CA, neoadjuvant CS/IL-12 significantly reduced the number of metastatic lung nodules when delivered before resection compared to resection alone (Figure 3F). Representative images of formalin-fixed lung tissue from each treatment group exhibit the typical level of metastasis, where examples of metastatic lung nodules are indicated with an arrow (Figure 3G).

Next, to examine the systemic effects of focal cryo-immunotherapy against established abscopal lesions, bilateral tumors were established and the larger of the two tumors was treated with cryo-immunotherapy when it measured between 150 and 500 mm^3^. While the vast majority of primary tumors treated with CA experienced complete regression in both the MC38 (Figure 4B) and MB49 (Figure 4E) models, there was minimal impact on the untreated contralateral tumor (Figure 4A,D). Furthermore, there was no benefit to overall survival with the addition of CS/IL-12 to CA (Figure 4C,F). These results were supported by similar outcomes in a bilateral Panc02 model (Appendix A). It should be noted that the tumor volumes of the untreated abscopal tumors ranged between 60 and 110 mm^3^ at the time of treatment, indicating that these tumors were more established and likely harder to control than the lung metastases in the LLC1 model.

### 3.4. CS/IL-12 Alters Differential Gene Expression in the Tumor Draining Lymph Node

Early attempts to explore immunophenotypic changes in tumor tissue after treatment via flow cytometry failed due to the overwhelming amount of necrotic and often liquefied tissue in the tumor bed that prevented viable cell collection. Nevertheless, enough RNA could be collected from tumor sites to perform transcriptome analyses. Thus, whole RNA sequencing on both tumor and dLN tissue at day 3 and day 7 post treatment with either CA alone or CA + CS/IL-12 was performed (Figure 5). Primary component analysis (PCA) using the entire available gene set of 67,228 genes revealed distinct clusters for tumor and dLN samples. However, the PCA values for tumor samples were more disperse than dLN samples revealing greater biological variation, with a predictive value of 73.77% for the primary component (Appendix A). Further, Pearson’s correlation analysis of tumor (Appendix A) and dLN (Appendix A) samples support the PCA results, indicating strong differences in gene expression between treatment groups.

In the dLN, the addition of CS/IL-12 to CA resulted in a major shift in gene expression both on day 3 (Figure 5B,D,F) and day 7 (Figure 5C,E,G) post treatment. On day 3 post treatment, an analysis of significant genes (Q < 0.05, CA alone vs. CA + CS/IL-12) revealed 637 differentially expressed genes (DEGs) which cluster into treatment groups after unbiased hierarchical clustering (Figure 5B). Upon further filtering with a fold change > 2 and Q < 0.05 cutoffs, there were 32 significantly upregulated genes, including Lef1, and 134 significantly downregulated genes, including Ifit3b, Ifit3, Cxcl16, and Il12b (Figure 5D).

As for gene expression in the dLN on day 7 post treatment, fewer genes were found to be significant with a total of 118 DEGs identified. However, these DEGs still clustered into treatment groups after unbiased hierarchical clustering, where 37 DEGs had a higher expression in the CA alone group and 81 DEGs had higher expression in the CA + CS/IL-12 group (Figure 5C). Few of these genes reached statistical significance after applying stricter filtering criteria with a fold change > 2 and Q < 0.05. Only 4 DEGs were found to be significantly upregulated (Gm46353, Gm10705, Fm14085, and Gabrr2) and 5 DEGs were significantly downregulated (Zfp965, Slc4a1, Lamb3, Arhgap24, and Plpp3) (Figure 5E). This may indicate a focusing of the immune landscape in the dLN over time.

By annotating significant DEGs on day 3 to the Kyoto Encyclopedia of Genes and Genomes (KEGG), we found that the top 25 KEGG pathways, based on the number of DEGs, included cell adhesion molecules (CAMs), metabolic pathways, antigen processing and presentation, cytokine–cytokine receptor interaction, and hematopoietic cell lineage (Figure 5F). Pathways that were unique to the dLN at day 3, indicated by the open bars, were endocytosis, NOD-like receptor signaling pathway, chemokine signaling pathway, and cellular senescence (Figure 5F). Nineteen of the top 25 KEGG pathways are shared between the day 3 and day 7 timepoints, indicated by solid bars, and 6 are unique to the day 7, indicated by open bars (Figure 5G). These changes indicate that a significant shift occurs in the dLN over time after treatment.

To compare how these pathways change over time, the immune pathways of interest, including antigen processing and presentation (orange), Th1 and Th2 cell differentiation (blue), and T cell receptor signaling pathway (green), and their rank in order was followed from day 3 to day 7 (Figure 5F,G). Antigen processing and presentation (orange) ranks third in the number of DEGs on day 3, but drops to 20th place by day 7. Meanwhile, B cell receptor signaling (purple), T cell receptor signaling (green), Th17 differentiation (red), and Th1 and Th2 cell differentiation (blue) all increase in rank over time to positions in the top 5 pathways (Figure 5H). To understand how these pathways are distributed between the treatment groups over time, the upregulated genes for CA alone and CA + CS/IL-12 (Figure 5B,C) were annotated to KEGG pathways separately and the top 10 pathways were ranked as before based on the number of DEGs belonging to that pathway (Appendix A). For CA alone, T cell receptor signaling pathways, Th17 cell differentiation, and Th1 and Th2 cell differentiation are in the top four pathways on day 3 and remain at the top on day 7, although the number of DEGs mapped to each pathway decreases (Appendix A). For CA + CS/IL-12, antigen processing and presentation ranks third on day 3, but then drops below the top 10 ranking on day 7 (Appendix A). Meanwhile, the B cell receptor signaling pathway ranks 8th on day 3, but then moves up to the first place position on day 7 (Appendix A). Taken together, these data point to the early role of CS/IL-12 inducing the first step of adaptive immunity after CA with antigen processing and presentation in the dLN. However, without CS/IL-12, there are more downstream pathways of adaptive immunity that are expressed early and sustained through day 7.

Within the tumor, a broad qualitative analysis of the DEGs on day 3 resulted in the identification of an outlier in the CA + CS/IL-12 group as well as high variability within the CA alone group. Difficulty in obtaining viable tissue from the highly necrotic tumor site after CA may have contributed to the poor quality. Tumor gene expression data from day 7 reveals that while the treatment groups are distinct from each other based on the DEGs present in the tumor, there are few immune-associated genes or pathways of significance (Appendix A).

When comparing treatment groups versus untreated controls on day 7, there is little change in the dLN after treatment with CA + CS/IL-12 (Appendix A) and only some significant change in the dLN with CA alone (Appendix A). As expected, the tumor data resulted in significant differences between both treatment groups and the untreated control (Appendix A).

### 3.5. CS/IL-12 Increases Gene Expression of Markers for Cross Presentation and Costimulation Early after CA

Further analysis of the key genes associated with dendritic cells (DCs), natural killer (NK) cells, and macrophages (Appendix A) revealed that Batf3, a key transcription factor for cross-presenting DCs, was significantly increased 3 days after CA + CS/IL-12 compared to CA alone in the dLN and trended higher early after CA in the tumor (Appendix A). Furthermore, Cd86 was significantly upregulated in the dLN and tumor on day 3, with the addition of CS/IL-12 to CA (Appendix A). Other trends demonstrating a possible mechanism for CS/IL-12 activity after CA included increased early levels of Klrk1 (NKG2D) in the dLN (Appendix A), and higher levels of Itgam (CD11b) in the tumor on day 3 (Appendix A). Overall, there was a trend toward higher levels of macrophage-associated (Itgam, Arg1, Nos2) and neutrophil-associated (Fcgr3) gene expression in tumors receiving CA or CS + CS/IL-12 compared to non-ablated tumors (Appendix A). However, with the exception of the day 3 Itgam expression in CA + CS/IL12 treated tumors, none of the other expression levels were statistically significant.

As for the expression of genes associated with T cells, general trends indicate higher levels of Gzma (Granzyme A) in the dLN on day 3 with CS/IL-12 compared to CA alone (Appendix A). However, both Cd4 and Cd8 expression in the dLN are higher with CA alone (Appendix A). These data may point to an initial explanation of mechanism where CS/IL-12 enhances antigen cross-presentation, co-stimulation, and subsequent T cell function through granzyme production.

### 3.6. Immune Suppression Increases over Time after CA Treatment

It has been demonstrated that IL-12, through the IFNg-IDO axis, can lead to regulatory T cell (Treg) rebound suppression and tumor escape [39,40]. In order to understand the regulatory response to treatment with or without IL-12 in combination with CA in the dLN, the expression of a subset of genes related to Tregs was analyzed and displayed as a heatmap (Figure 6A). Unbiased hierarchical clustering resulted in day 3 samples being separated from day 7 samples; however, the type of treatment was not easily differentiated (Figure 6A). Genes particularly upregulated from day 3 to day 7 in CS/IL-12 treated mice are Rel, Ptprc, and Entpd1 (Figure 6A). Both Rel and Ptprc, which encode an NF-kB subunit and CD45, respectively, play a role in lymphocyte survival and proliferation, while Entpd1 is an ATPDase and is implicated in pro-tumorigenic activity [41,42,43]. Foxp3 expression, a transcription factor associated with Tregs, was low on day 3 and then increased over time in both treatment groups in dLNs (Figure 6B) and tumors (Figure 6C). However, the level of Foxp3 expression in the tumor was lower at day 7 for the CA + CS/IL-12 group compared to CA alone (Figure 6C). This may indicate a temporary suppression of Treg activity in the tumor by IL-12. As expected, the level of Ifng expression was the highest 3 days after CA + CS/IL-12 treatment, but this expression was not sustained, dropping down to CA alone levels by day 7 (Figure 6D). Ido1 expression mirrored this trend, though the levels were not statistically significant (Figure 6E).

The addition of CS/IL-12 to CA did not have an effect on the Pd1 expression in the dLN compared to CA alone, which increased over time and at the same levels for both treatment groups, returning to untreated tumor levels of expression by day 7 (Figure 6G). However, Pdl1 expression in the tumor was significantly higher in the CA + CS/IL-12 treatment group on day 3 compared to all other days and treatment groups (Figure 6F). Analyzing other coinhibitory markers, Ctla4, Tigit, and Tim3 expression in the dLN was not different between the two treatment groups, but expression levels did increase over time from day 3 to day 7 (Figure 6H–J). Altogether, these data indicate that immune suppression and regulation increases over time in the dLN after treatment with CA, and that there may be an immunosuppressive rebound response to CS/IL-12.

### 3.7. Addition of Checkpoint Inhibitors to CA plus CS/IL-12 Delays Abscopal Tumor Growth

Given the high levels of gene expression in the dLN for Pd1, an immune checkpoint, we evaluated the effect of anti-PD-1 therapy in combination with CA and intratumoral CS/IL-12. Immune checkpoint inhibitor therapy in combination with ablation has demonstrated abscopal tumor regression in multiple preclinical models with small (<200 mm^3^) tumors [44,45]. To evaluate the potential synergistic effect of checkpoint inhibitor therapy with CA + CS/IL-12, MC38 bilateral tumors were established and the larger of the two tumors was treated 10 days after implantation (Figure 7A). The addition of anti-PD-1 to CA + CS/IL-12 prolonged median survival from 16 to 24 days compared to isotype + CA + CS/IL-12 (Figure 7B). The survival curves that were significantly different from the triple combination therapy were the isotype control (*p* = 0.0003), anti-PD-1 control (*p* = 0.0002), and anti-PD-1 + CA group (*p* = 0.0018). Though not statistically significant (*p* = 0.1488), it is worth noting that the isotype + CA group contained 1 mouse out of 7 total that experienced an abnormally long survival compared to the rest of the treatment group (Figure 7D).

Additionally, the triple combination significantly delayed abscopal tumor growth up to 500 mm^3^ compared to all other treatment groups (Figure 7C). This delay in growth is further demonstrated by the tumor growth curves (Figure 7D). Where antibody-only treatments do not induce tumor regression, anti-PD-1 therapy in addition to CA does slightly delay tumor growth, thus indicating the mild sensitivity of MC38 to anti-PD-1 treatment (Figure 7D). The addition of anti-PD-1 to CA + CS/IL-12 delays abscopal tumor growth even further; however, the large abscopal tumor volumes (>200 mm^3^) and established immunosuppressive microenvironments at the time of treatment most likely limit the expected impact of anti-PD-1, as shown in other studies using much smaller tumors (Figure 7D) [46]. In summary, these data demonstrate that the systemic antitumor immunity induced by CA and localized CS/IL-12 can be modestly improved with a systemic immunotherapy, such as anti-PD-1. However, the results also reveal that large tumors with an established suppressive microenvironment are difficult to treat in an abscopal setting.

## 4. Discussion

Minimally-invasive tumor ablation technologies, including CA, feature low complication rates, reduced recovery times and lower overall hospitalization costs compared to surgical resection. Unfortunately, high rates of recurrence following ablation of large tumors prevent their widespread clinical use. Thus, the initial goal of our research was to determine if a localized immunotherapy could prevent recurrences of large tumors after CA. Our previous research demonstrated that intratumoral injections of CS/IL-12 were effective in eliminating small (~50–100 mm^3^) tumors [30,33]. Given the recent clinical translation of several localized IL-12 immunotherapies [47,48,49,50], CS/IL-12 was a logical candidate to combine with CA. To our knowledge, no other localized IL-12 delivery system has been used as an adjuvant to CA. Systemic administration of IL-12 after CA was shown by others to increase the number of DCs in the draining lymph nodes, tumors, and spleens of rats bearing C6 gliomas; however, this combination did not improve antitumor activity compared to CA alone [17].

In this study, we demonstrated that CS/IL-12 improved the prevention of primary tumor recurrence after percutaneous CA compared to two other immune agonists, Resiquimod and DMXAA, (Figure 1) which have shown antitumor activity in multiple murine models [51,52,53,54]. It should be noted that the delivery of these agonists was not optimized, and we are currently working on a sustained release platform for small molecule immunomodulators. Nevertheless, we found that adjuvant CS/IL-12 prevented primary recurrences following CA of large (200–500 mm^3^) tumors in multiple models, including MB49, MC38, Panc02, and even in the non-immunogenic B16.F10 model. Given our previous demonstrations that localized CS/IL-12 is safe and results in limited systemic dissemination of IL-12 [34,55], this cryo-immunotherapy combination can be considered for clinical translation against diverse solid tumors. Practically speaking, clinical translation of adjuvant localized IL-12 might be most efficient in solid cancers for which CA is already used, such as kidney, prostate, breast, lung and bone cancers. As an example, adjuvant CS/IL-12 could be explored in larger (>4 cm) renal masses for which CA is not recommended due to higher recurrence rates compared to partial nephrectomy [20].

When administered in the neoadjuvant setting, CS/IL-12 eliminated spontaneous lung metastases in most animals, following CA (Figure 3). However, against established abscopal tumors, CA + CS/IL-12 had limited impact (Figure 4). The addition of systemic anti-PD-1 resulted in measurable tumor growth delay; however, regression of untreated tumors was not achieved (Figure 7). These data are in contrast to our previous research showing that localized CS/IL-12 alone induced robust protective immunity and complete abscopal responses [30,56]. It should be noted that these prior studies treated much smaller tumors and with multiple intratumoral injections. Thus, it is possible that multiple CS/IL-12 injections are needed to boost systemic antitumor immunity. It may also be the case that the injury and inflammation induced by CA dampens antitumor immunity.

Relative to other published cryo-immunotherapies, CA + CS/IL-12 compares favorably. For instance, intratumoral injection of both immature DCs and CpG after CA of D122 tumors measuring 12–20 mm^3^, resulted in an overall survival of 67% at 120 days post tumor inoculation [57]. Primary tumor growth in the CA plus, both CpG and DC group, was also significantly slowed compared to CA with either CpG or DC [57]. Another group, also using peritumoral injection of DCs and CpG after CA, demonstrated extended survival, though no durable cures in a s.c. LLC model with tumor measuring 200–220 mm^3^ at the time of treatment [58]. In the CT26 s.c. model, primary tumors grown to 14 days post inoculation and measuring approximately 180 mm^3^, were treated with CA followed by intratumoral injection of DCs pretreated with Bacillus Calmette-Guerin cell wall skeleton (BCG-CWS). This cryo-immunotherapy significantly slowed the growth of the primary tumor as well as the growth of a secondary untreated tumor on the opposite flank compared to intratumoral injection of immature DCs after CA [59]. However, it is not clear if any of primary tumors were eliminated or if any of the mice had survived long-term.

The results of our transcriptome analyses indicate potential mechanisms by which adjuvant CS/IL-12 prevents primary tumor recurrence following CA. Differential gene expression data revealed that the addition of CS/IL-12 to CA caused a significant shift in gene expression in the dLN and these differences persisted for at least 7 days post treatment (Figure 5). An increase in IFIT genes, including Ifit3b and Ifit3, at day 3 with the addition of CS/IL-12 points to a shift in immune function, although the exact antitumor functions of IFIT genes are still debated [60]. The genes for the IL-12 subunit p40 (Il12b) and the chemokine CXCL16 are also significantly upregulated on day 3 in the dLN after treatment with CA + CS/IL-12 (Figure 6G). IL-12 has strong antitumor properties mediated through NK and cytotoxic T cells [61]. CXCL16 is induced by IFNγ and TNFα and serves as a chemotactic signal for activated T cells [62]. We also classified these significant genes to KEGG pathways and found that while many immune-related pathways ranked high early after treatment (Figure 6F), T cell differentiation and receptor signaling genes were upregulated by CA alone, while antigen presentation and B cell signaling genes were upregulated following CA + CS/IL-12 (Appendix A). On day 7 after treatment, these T cell associated pathways remained high on the rank list for CA alone, but antigen processing and presentation fell below the top 10 ranked pathways for CA + CS/IL-12 (Appendix A). These data indicate that CS/IL-12 may enhance antigen presentation early, but the resultant long-term T cell response appears limited. A potential limitation of the gene expression studies is that only one model, MC38, was explored. Although similar findings would be anticipated in other models that exhibited similar antitumor responses (Figure 1), additional experiments would be required to confirm the actual findings.

The injury and inflammation induced by CA may dampen antitumor immunity through multiple mechanisms of regulatory rebound. The high level of Pdl1 expression in the tumor on day 3 after CA + CS/IL-12 treatment (Figure 6G) indicates a suppressive tumor microenvironment, which may prevent robust T cell activation and subsequent development of memory T cells [63,64,65]. However, the addition of anti-PD-1 to CA + CS/IL-12 serves only to delay tumor growth in the bilateral MC38 tumor model (Figure 7). Another explanation may be the high levels of IFNg produced after CS/IL-12 delivery. IFNg has been linked to an increase in IDO expression on DCs and the subsequent activation and proliferation of Tregs [39,40]. Expression levels of Ifng, and Ido1, to a lesser extent, peak early at day 3 after CS/IL-12 delivery, but remain low after treatment with CA alone (Figure 6D,E). Further research is necessary to solidify this potential mechanism. Regardless of treatment, the increase in expression of immune regulatory genes, such as Foxp3, Ctla4, Tigit, and Tim3 over time may indicate that there is a regulatory rebound to CA therapy (Figure 6B,C,H–J). This regulatory rebound is not affected by the subsequent delivery of CS/IL-12 after CA. These hypotheses are the subject of ongoing and future work.

## 5. Conclusions

In conclusion, this study demonstrates that the addition of intratumoral CS/IL-12 immunotherapy to CA improves the prevention of large primary tumor recurrence after focal ablation. Localized CS/IL-12 overcomes the safety limitations associated with systemic IL-12 delivery and is thus deserving of clinical consideration as an adjuvant immunotherapy. In terms of systemic immunity, CS/IL-12 prevented lung metastases when delivered as a neoadjuvant to surgery or CA. However, CS/IL-12 was not able to slow abscopal tumor growth after CA in bilateral tumor models. Abscopal tumors grew statistically slower when anti-PD-1 was added to CA + CS/IL-12, but there were no durable regressions of the untreated tumor. While the RNAseq analysis gives some initial hints at the mechanisms behind these findings, future in-depth studies are necessary to fully elucidate the underlying processes of the antitumor immunity induced by the addition of CS/IL-12 to CA.

## Figures and Tables

**Figure 1 cancers-15-02210-f001:**
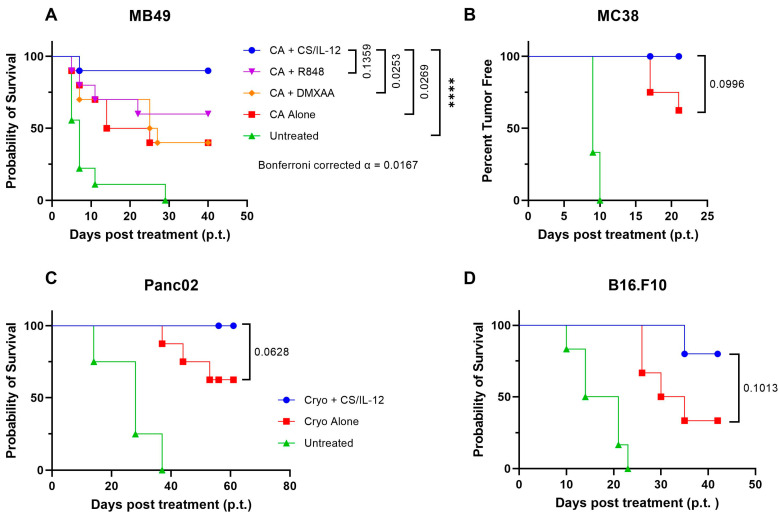
CS/IL-12 improves primary tumor elimination after CA in multiple models. (**A**) In the MB49 model (n = 10), mice were treated with CA and either a TLR7/8 agonist (R848), a STING agonist (DMXAA), or CS/IL-12 two days post CA. A table of *p*-values for the comparison of treatment groups using the Log-rank test is provided. In all other models (n = 8), including MC38 (**B**), Panc02 (**C**), and B16.F10 (**D**), CS/IL-12 was delivered intratumorally (i.t.) on the same day within an hour of CA. Tumor volume and survival was monitored and the threshold for euthanasia was 2000 mm^3^. Statistical significance was calculated using the Log-rank (Mantel–Cox) test. **** *p* < 0.0001.

**Figure 2 cancers-15-02210-f002:**
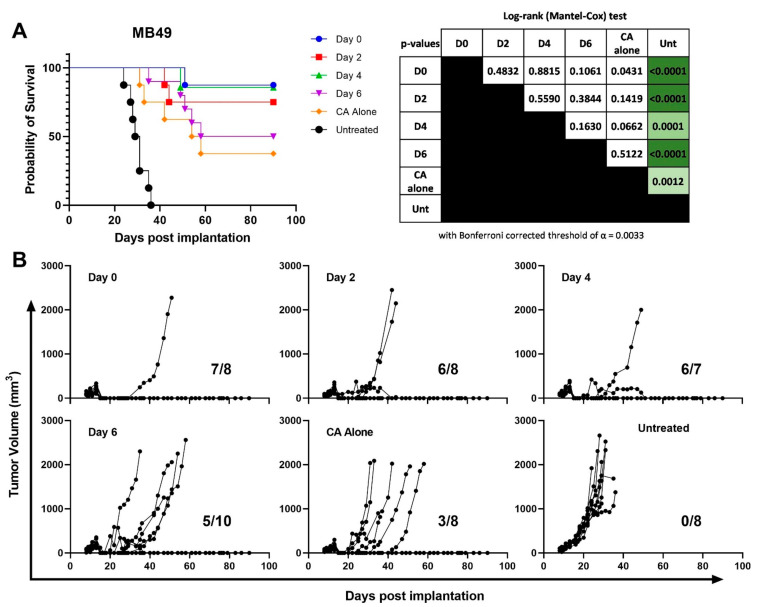
Effect of adjuvant CS/IL-12 timing on recurrence prevention. In the MB49 model, mice were treated with either CA alone or CA + CS/IL-12 delivered either within an hour of CA (Day 0), 2 days, 4 days, or 6 days after CA. Survival (**A**) and tumor volume (**B**) were monitored for 90 days post implantation. Statistical significance was calculated using the Log-rank (Mantel–Cox) test and a table of *p*-values for each comparison is shown next to the survival curve (**A**). Comparisons with a significant difference below the Bonferroni corrected threshold are highlighted in green. Fractions in (**B**) refer to the number of mice without a primary tumor recurrence at the end of the study.

**Figure 3 cancers-15-02210-f003:**
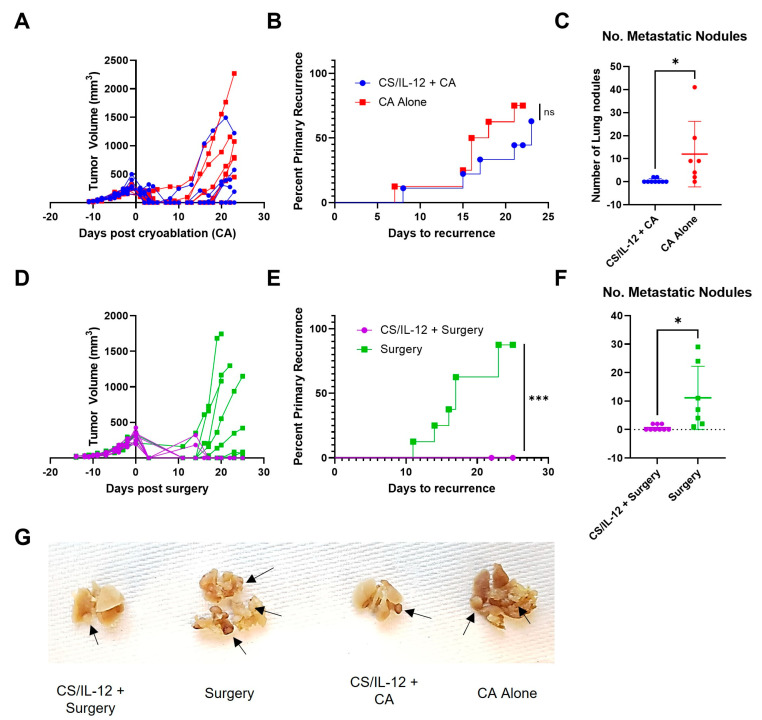
Addition of neoadjuvant CS/IL-12 to CA and tumor resection inhibits lung metastases. Lewis lung carcinoma (LLC1) cells were implanted subcutaneously in the right flank and treated with either CA alone (n = 9) or CA + CS/IL-12 delivered intratumorally 2 days before ablation (n = 9) (**A**–**C**) or tumors were resected with (n = 10) or without (n = 8) CS/IL-12 neoadjuvant (**D**–**F**). Tumor volume (**A**,**D**) was monitored and the number of days to recurrence (**B**,**E**) was measured as the first day post CA that a tumor mass was measurable. All mice were euthanized on either day 22 or 23 after CA and lungs were dissected in order to visualize and count metastatic nodules (**C**,**F**). Representative lungs from each group are shown in (**G**), where examples of metastatic nodules are indicated with an arrow. Statistical significance was calculated using the Log-rank (Mantel–Cox) test (**B**,**E**) and Student’s *t*-test (**C**,**F**). ns—not significant; * *p* < 0.05; *** *p* < 0.001.

**Figure 4 cancers-15-02210-f004:**
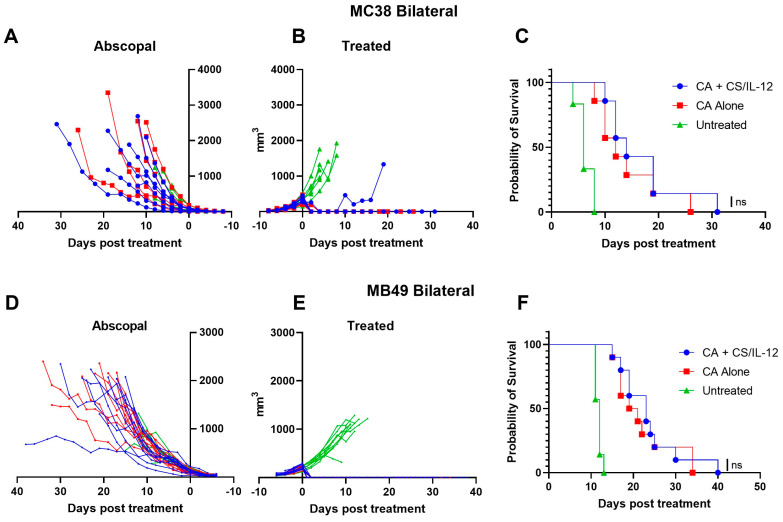
Addition of CS/IL-12 to CA does not slow established abscopal tumor growth. In both the MC38 (n = 8) and MB49 (n = 10) models, 3 × 10^5^ and 1.5 × 10^5^ cells were subcutaneously implanted on the right and left flanks, respectively. The larger of the two tumors was treated with CA alone or CA + CS/IL-12, administered immediately post-CA, upon reaching 150–500 mm^3^. Individual tumor volumes on untreated (**A**,**D**) and untreated (**B**,**E**) sides were measured and survival (**C**,**F**) was monitored. Statistical significance was calculated using the Log-rank (Mantel–Cox) test (**C**,**F**). ns—not significant.

**Figure 5 cancers-15-02210-f005:**
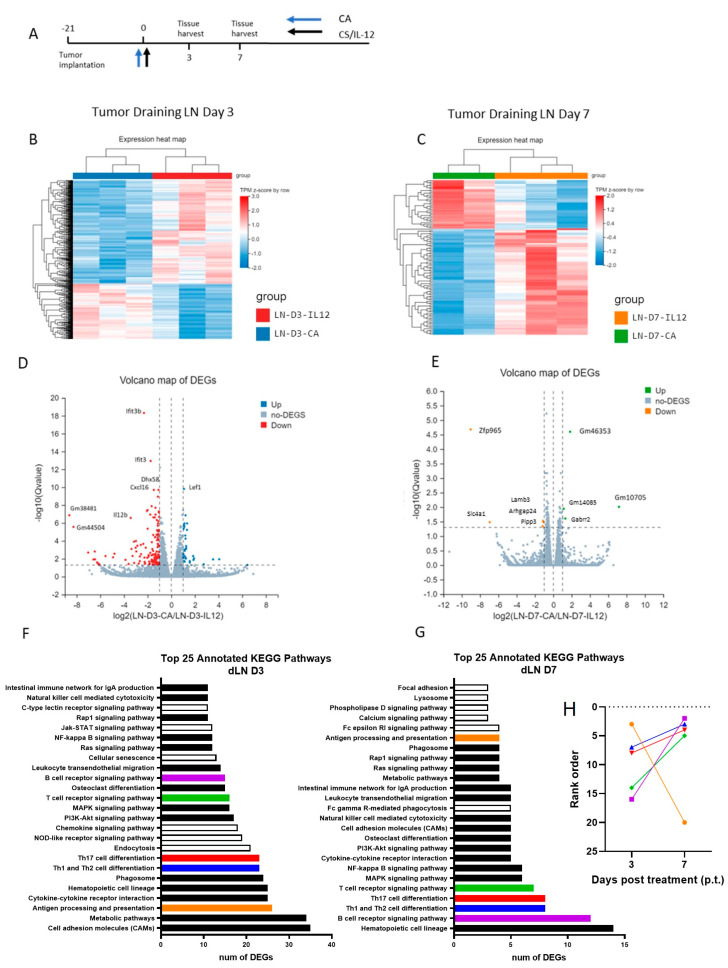
CS/IL-12 alters gene expression in the tumor draining lymph node. (**A**) C57BL/6 female mice were implanted with a single MC38 tumor and were treated with either CA alone or CA + CS/IL-12 and total RNA was extracted from the draining inguinal lymph node on day 3 (**B**,**D**,**F**) or day 7 (**C**,**E**,**G**) after treatment. All genes with Q < 0.05 were visualized with heatmaps for day 3 (**B**) and day 7 (**C**). Volcano plots, where the threshold for significant difference was |log2FC| ≥ 1 and Qvalue ≤ 0.05 (**C**,**D**). Gene annotation was performed using the Kyoto Encyclopedia of Genes and Genomes (KEGG) and the top 25 most frequent pathways were plotted (**F**,**G**). Five immune pathways of interest shared between day 3 (**F**) and day 7 (**G**) were color-coded in order to follow their ranking over time. A rank of 1 indicates the highest number of DEGs in that set. The change in rank is plotted as a line graph in (**H**) where the color code is maintained. Black bars are other pathways shared between day 3 and day 7. Open bars are pathways that are unique to that analysis time point.

**Figure 6 cancers-15-02210-f006:**
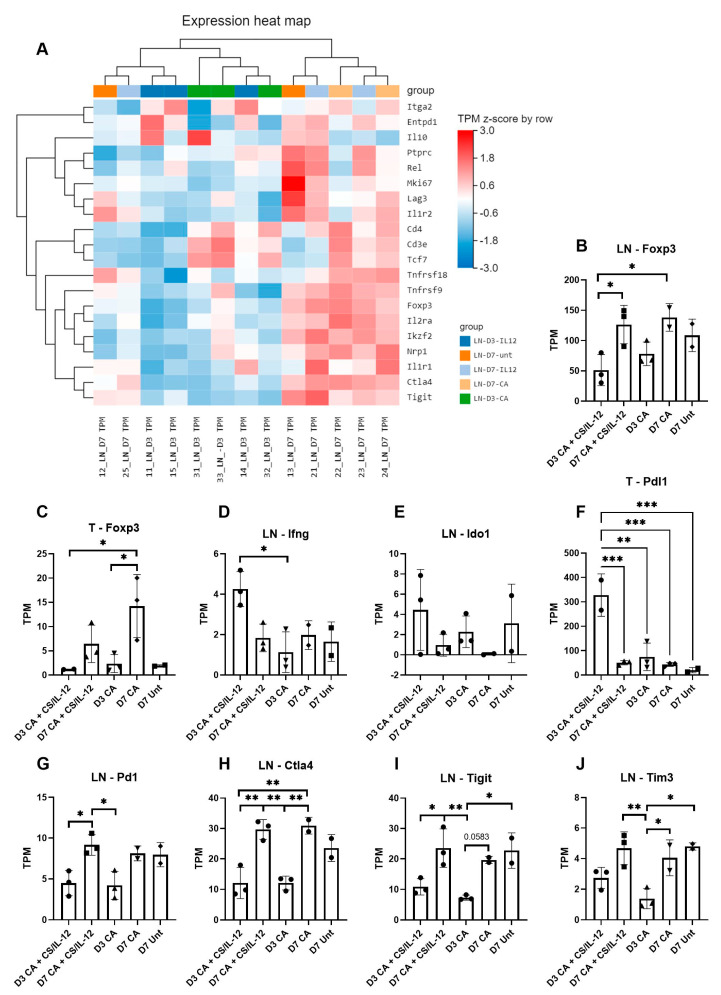
Immune regulation increases over time. Mice bearing MC38 tumors were treated as indicated and then tissue samples were acquired on day 3 (D3) or day 7 (D7) post treatment. The unit of measure is transcripts per kilobase million (TPM) using z-score (row direction) as the standardization method and the horizontal axis was sorted based on cluster order. Gene expression of a subset of genes associated with regulatory T cells was quantified as TPM and standardized using log2 (value + 1). The results are displayed as a heatmap using all LN samples (**A**). A quantitative comparison of selected genes was performed based on TPM values (**B**–**J**). Foxp3 was assessed in both the LN (**B**) and tumor (**C**) samples, Ifng (**D**) and Ido1 (**E**) were analyzed in the LN samples, and Pdl1 in the tumor (**F**) and Pd1 in the LN (**G**). The immune checkpoint Ctla4 (**H**), Tigit (**I**), and Tim3 (**J**) were all assessed in the LN. Each point represents a biological replicate. Statistical significance was calculated using a one-way analysis of variance with Tukey multiple comparison correction. Ns—not significant; * *p* < 0.05; ** *p* < 0.01; *** *p* < 0.001. Abbreviations: LN, tumor draining lymph node; D3, day 3 after treatment; D7, day 7 after treatment; Unt, untreated tumor.

**Figure 7 cancers-15-02210-f007:**
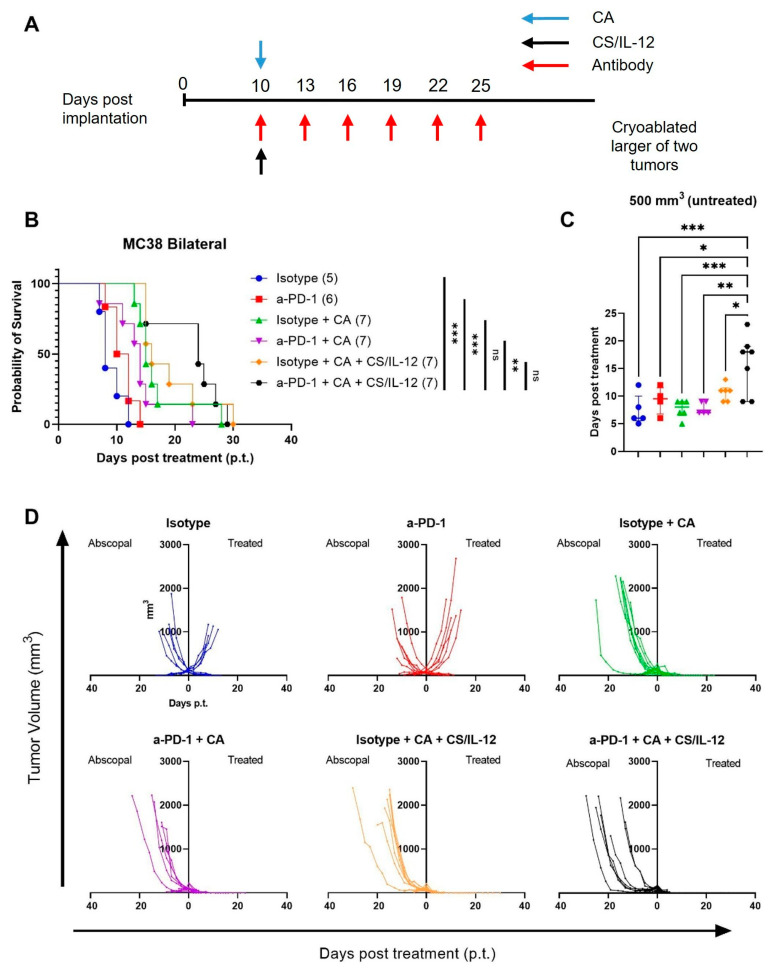
Addition of checkpoint inhibitors to CA + CS/IL-12 delays abscopal tumor growth. In the MC38 model, bilateral tumors were implanted on the right and left flanks and the larger of the two tumors was treated when it reached between 150 and 200 mm^3^. (**A**) Antibody treatment (300 μg) with either anti-PD-1 or isotype control began on the same day as CA and was delivered i.p. every 3 days for a total of 6 doses. Survival (**B**) and tumor volume (**D**) were monitored for 30 days post treatment. The number of days post treatment for the abscopal tumor to reach 500 mm^3^ (**C**) was calculated as the first day the contralateral tumor measured ≥ 500 mm^3^. The number in parentheses next to the legend label for each group indicates the number of mice in that treatment group (**B**). Statistical significance was calculated using the Log-rank (Mantel–Cox) test (**B**) and a one-way analysis of variance with Tukey multiple comparison correction (**C**). ns—not significant; * *p* < 0.05; ** *p* < 0.01; *** *p* < 0.001.

## Data Availability

Gene expression data that have been cleaned (QC and Filter) and aligned to the reference genome have been deposited on the Dryad Data Platform and are available at: https://doi.org/10.5061/dryad.g4f4qrfts. All other data are available on request.

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
