# Peer review of "Focal Cryo-Immunotherapy with Intratumoral IL-12 Prevents Recurrence of Large Murine Tumors"

_cancers, 2023, doi:10.3390/cancers15082210_

Round 1

Reviewer 1 Report

The article titled “Focal cryo-immunotherapy with intratumoral IL-12 prevents recurrence of large murine tumors” describes a combinatory approach using cryoablation (CA) and IL-12 in preclinical models. The reviewer is impressed by the variety of cancer models (MB49, MC38, Panc02, LLC1, B16 etc.). The reviewer also greatly appreciates different approaches of assessing the benefits of cryoablation + IL-12, including primary tumor, lung metastasis, abscopal effect and gene expression of the dLN.

However, the reviewer would raise a few major concerns to be addressed before recommending for publication.

1.        Lack of consistency of cryoablation

The reviewer is an active user of cryoablation systems. I agree with the authors’ argument that “Residual cancer deposits following CA drive higher rates of local recurrence…” and that “The injury and inflammation induced by CA may dampen antitumor immunity”. Therefore, it is never trivial to have a well-controlled cryoablation procedure on the tumor so that studies are comparable and repeatable, as how a tumor is being treated has a profound effect in immunity.

In this article, different cryoablation procedures are being used, for instance:

·         Caption of Fig 4: CA + CS/IL-12 when the tumor reached between 150 – 200 mm3.

·         Page 9: cryo-immunotherapy when it measured between 200-400 mm3

·         Page 5: large (200-500 mm3) murine tumors were established subcutaneously on the flank and treated with either CA alone or CA plus intratumoral immunotherapy

·         2021 presentation titled “Onco-immunological mechanisms of focal ablation and localized IL-12 immunotherapy.”: MC38 tumors were treated with a minimal cryoablation protocol where the tumor undergoes one cycle of freezing at 100% intensity up to the tumor margin by visual inspection, followed by one cycle of active thaw until the cryo probe can be removed.

·         Page 3: three freeze/thaw cycles with 1 min at 100% freezing intensity and then 2 min of active thaw

Given the variety of tumor size while keeping the freezing time constant (1 min) and different freeze/thaw cycle, it appears that the prescribed “dosing” of cryoablation is different across the board.

2.       Lack of proper control

In the gene sequencing assessment, only 2 groups are compared: CA vs. CA/IL-12. There is no IL-12 alone and non-treated.

3.       Lack of in-depth analysis mechanism of action

·         The RNAseq analysis is limited to only the MC38 model, whose cryoablation procedure may be different (see comment #1) to other procedures.

4.       RNA sequencing of tumor samples.

The reviewer is not convinced that there is sizable viable cancer cells or tumor residue left on Day 3 and 7 after cryoablation, to yield any meaningful results. Most of the collected cells could be infiltrated macrophage, neutrophil and other cells than the surviving cancer cells. RNA degradation is measured by hours (or faction of hours) at body temperature.

·         The authors describe that “flow cytometry failed due to the overwhelming amount of necrotic, often liquefied, tissue in the tumor bed that prevented viable cell collection”

·         “Total RNA was extracted from the cell suspension”

·         Fig 1B, percentage of tumor-free is 100% until day 17 post-treatment.

·         Fig 4B, tumor volume is “0” before day 9.

Other minor concerns and questions

1.       What is the timing of CS/IL-12 for the abscopal model?

2.       Fig 6, some parts are impossible to read

3.       Please explain the inconsistency of the source of animals.

·         In this paper page 2: Female C57BL/6 mice aged 6-12 weeks old were obtained from Charles River Laboratories

·         In the 2020 presentation titled “Intratumoral interleukin-12 administered after cryoablation does not improve survival in multiple bilateral murine models.”: Female C57BL/6 mice were purchased from Jackson Laboratory.

4.       Inconsistency of anti-PD-1 timing

Page 3, “For the anti-PD-1 and isotype antibody (BioXCell, clone: RMP1.14) treatments, 300 μg was injected i.p. every 3 days starting on the day of CA for a total of 4 doses” does not match the timeline presented Fig 7A.

5.       Why wasn’t flow cytometry performed on dLN, spleen or blood? 

Author Response

Thank you for your thoughtful and thorough review. We appreciate your expertise in this field as well as the opportunity to strengthen our manuscript by addressing your concerns.  Regarding the consistency of cryoablation (critique #1), all procedures in this manuscript utilized 3 freeze-thaw cycles. In all instances, despite some differences in tumor size, the iceball encapsulated the entire tumor for complete cryoablation. This information is added in the revised manuscript (lines 108-109). None of the data from the 2021 presentation which used only 1 freeze-thaw cycle are reported here. In the previous 2021 presentation we were trying to establish a 100% recurrence model which is not relevant here. Regarding the size of primary tumors treated, our goal was to use tumors between 200-400mm3. Because some tumors grew faster than anticipated, we expanded the range to 200-500mm3 for some experiments. The studies in Figure 4 were performed in mice bearing bilateral tumors rather than solitary, primary tumors. Because of the additional tumor burden the minimum target tumor volume for treatment was reduced to 150mm3 with a maximum acceptable volume of 500mm3. The caption for Figure 4 has been adjusted in the revised manuscript.

With respect to the control groups in RNAseq studies (critique #2), we did have a non-treated control group at the day 7 timepoint (see Figure 6). We did not use a non-treated control group at day 3 timepoint as minimal changes would be expected in 4 days in the absence of treatment. Also, we chose not to include an IL-12 alone treatment as the primary goal of these studies was to determine how CS/IL-12 enhances CA and not how CS/IL-12 alone compares to IL-12 alone.

Regarding the lack of in-depth mechanism of action studies (critique #3), we explored only the MC38 model as all primary tumor models performed similarly. However, the MC38 model was treated with 3 freeze-thaw cycles like all models in this study. The limitation of performing RNAseq on only one model is noted in the revised manuscript (lines 582-585). We have chosen not to pursue additional mechanistic studies at this time as we are in the process of developing a novel, tunable chitosan hydrogel that appears to be more effective at sustaining IL-12 in the tumor following injection and demonstrates greater antitumor activity than the CS formulation used in this manuscript. Rather than performing complete mechanistic studies using our older CS formulation, we would like to publish the current studies in a timely manner and focus on mechanistic studies using our next generation hydrogel once this new system has been completely characterized.

On the RNA sequencing of tumor samples (critique #4), an attractive feature of whole transcriptome sequencing is that it requires far fewer cells than flow cytometry. Properly controlled flow cytometry requires at least 100,000 cells whereas RNAseq can be performed with 10,000 cells.  We agree with the reviewer that the collected cells could include immune infiltrates. Our analyses indicate that macrophages, neutrophils and lymphocytes are indeed present within ablated and non-ablated tumors (see Figures S6, S7). And overall, there was a trend toward higher levels of macrophage-associated (Itgam, Arg1, Nos2) and neutrophil-associated (Fcgr3) gene expression in tumors receiving CA compared to non-ablated tumors (Fig S6 J, L, N, P) as one might expect. However, with the exception of the day 3 Itgam expression in CA + CS/IL12 treated tumors, none of the other expression levels were statistically significant. This information has been included in the revised manuscript (lines 414-418).

As for the minor concerns and questions, 1) the timing of CS/IL-12 for the abscopal model has been included in the revised manuscript (line 316-317); 2) a higher resolution version of Figure 6 has been uploaded; 3) all animals used to collect data from this manuscript were indeed purchased from Charles River although we see no differences in tumor growth or treatment in mice from different vendors; any prior references to the use of mice from Jackson Labs are inaccurate; 4) the anti-PD-1 administration schedule has been clarified (lines 139), specifically 6 doses were administered as shown in Figure 7A; 5) once it was determined that flow cytometry could not be performed on tumor samples, a decision was made to pivot to RNA seq. However, this is a fair point and we will pursue flow cytometry in future studies involving the combination of CA with our IL-12 / hydrogel formulation.

Once again, we appreciate your rare expertise and hope there is an opportunity to collaborate in the future.

Reviewer 2 Report

The overall intention of the submission is scientifically sound, and the manuscript is very well written. The authors' hypothesis and the effect of CS/IL-12 on several tumors are promising, especially in terms of preventing lung metastasis. The addition of intratumoral CS/IL-12 immunotherapy to treat tumors followed by cryoablation greatly benefits the prevention of tumor recurrence and improves the survival rate.

The manuscript provides transcriptomic data that sheds light on the mechanism behind the CS/IL-12 treatment's anti-tumor immunity. However, further studies are needed to fully elucidate the underlying processes behind this to implement this approach in a clinical setting.

In addition, the authors should discuss what type of tumors may benefit more from this approach, as cryoablation is used to treat a variety of solid tumors, such as breast and bone tumors. It is important to determine whether the combination of CA+CS/IL-12 would show an effective response in all solid tumors.

Before accepting the paper, the authors need to revise the Table in Figure 1 as the p-values and the studied subjects were misplaced. Additionally, Figure 6 is not in high-resolution, making it difficult to read the expression heat map. The authors should revise this figure and submit a high-resolution version.

In summary, the manuscript is promising and well-written. However, some revisions are needed before it can be accepted for publication.

Author Response

Thank you for your positive review. We agree that additional experiments are needed to fully elucidate the immunological mechanisms of CA + CS/IL-12. We have chosen not to pursue additional mechanistic studies at this time as we are in the process of developing a novel, tunable chitosan hydrogel that appears to be more effective at sustaining IL-12 in the tumor following injection and demonstrates greater antitumor activity than the CS formulation used in this manuscript. Rather than performing complete mechanistic studies using our older CS formulation, we would like to publish the current studies in a timely manner and focus on mechanistic studies using our next generation hydrogel once this new system has been completely characterized.

In terms of tumor types that may benefit from cryo-immunotherapy, our intention with Figure 1 was to show that CA+CS/IL-12 is effective against diverse solid tumor models including both immunogenic (Panc02, MC38) and non-immunogenic (B16F10) tumors. While we believe that CA + CS/IL-12 would be effective in all solid tumors, in reality, clinical translation is expected first in cancers cryoablation already has a foot hold, such as in the management of kidney, breast, lung and bone cancers. The addition of adjuvant immunotherapy is expected to reduce recurrence rates and improve clinical outcomes for these patients. A discussion of these points has been included in the revised manuscript (lines 535-539).

Figures 1 and 6 have been revised as suggested for readability.

Reviewer 3 Report

The study evaluates the efficacy of localized immunotherapy with CS/IL-12 in preventing tumor recurrence after cryoablation (CA) in murine tumor models, and finds that it reduces recurrence rates by 30-55% and induces complete durable regression of large tumors in 80-100% of treated animals. The combination therapy also has limited systemic antitumor immunity.

There are some questions need to be addressed.

1.       It would be more meaningful if the author highlights the meaning of CS/IL12, and show how much IL12 can be capsulated in the CS.

2.       Have the authors tried direct intravenous or orthotopic injection of IL-12?

3.       How many days can CS/IL-12 stay in the tumor?

Author Response

Thank you for your time and effort in reviewing our manuscript. We have added additional details about CS/IL-12 in the revised manuscript (lines 73-77). Our previous description of CS/IL-12 is also included in the Methods section (lines 133-135). Importantly, we are using a solution of CS in these studies, and thus, IL-12 is not encapsulated in chitosan particles, but rather co-formulated in a chitosan solution. We have not attempted intravenous injection of IL-12 as systemic delivery of this potent cytokine is known to induce significant toxicity. However, we have performed orthotopic injections of CS/IL-12 in a mammary adenocarcinoma model (ref [34]). There do not appear to be any limitations on the types or locations of solid tumors that can be injected with CS/IL-12 as long as ultrasound or CT guidance is available. We have previously shown that CS/IL-12 can be maintained in the tumor for at least a week. This information has also been included in the revised manuscript (Lines 73-77).  

Reviewer 4 Report

The objective of this research was to determine if localized immunotherapy with CS/IL-12 could prevent tumor recurrence after cryoablation (CA). 

The aim is interesting and some amendments are required before publication:

  • CA is an established method for treating small renal masses. This is true and recent evidence shows how this is not only an alternative to surgery but an equivalent statement even in challenging indications. Please cite the following article on the topic DOI: 10.1016/j.ejso.2022.09.022.
  • I advice to shorten the conclusions.
  • The ablative technique has been proposed for different tumors. The techniques include radiofrequency ablation, microwave ablation, and cryoablation, utilize diverse forms of energy and physical mechanisms to directly kill tumor tissues. A recent paper reported a comparison among those which should be included in your article DOI: 10.23736/S2724-6051.22.05092-3.
  • Check typos 

Author Response

Thank you for your consideration of our manuscript. We have clarified that CA is an established method for treating small renal masses (lines 56-58) and the provided article has been cited (ref [20]) in the revised manuscript. In addition, the recent paper comparing different ablative techniques has been cited (ref [10]). Typos and grammatical errors have been thoroughly investigated.

Round 2

Reviewer 1 Report

The reviewer greatly appreciates the effort by authors in addressing my previous comments.

Regarding the CA procedure: the statements of "iceball encapsulated the entire tumor (volumes measuring between 200-500 mm3)" and "three freeze/thaw cycles with 1 min at 100% freezing intensity and then 2 min of active thaw" cannot be true simultaneously. Larger tumors take substantially more time to freeze till the edge of the tumor than smaller tumors. 

Author Response

Thank you for your efforts in reviewing our revised manuscript. We agree that larger tumors will take more time to freeze, however, even the largest tumors in our study (~500mm3) were visibly frozen after 1 minute of 100% intensity freeze. So the statement that the “iceball encapsulated the entire tumor” remains true.  The trade-off is that the 1min freeze /2min thaw x 3 cycles was ‘overkill’ for the smaller tumors. We could have used 20-40 second freeze periods for smaller tumors, however, we wanted to fix the treatment dose in this manuscript. Moving forward, we will adjust the times of freezing/thawing for each tumor. This approach would be more clinically relevant.  

Reviewer 3 Report

This manuscript can be published with this version.

Author Response

Thank you again for taking the time to consider our revised manuscript.

Reviewer 4 Report

A deeply improved revised version has been provided. In my opinion, the manuscript is worthy of publication in its current form.

Author Response

Thanks, we appreciate your effort in re-reviewing our manuscript.